# ECG Indices Poorly Predict Left Ventricular Hypertrophy and Are Applicable Only in Individuals with Low Cardiovascular Risk

**DOI:** 10.3390/jcm9051364

**Published:** 2020-05-06

**Authors:** Małgorzata Chlabicz, Jacek Jamiołkowski, Marlena Paniczko, Paweł Sowa, Małgorzata Szpakowicz, Magda Łapińska, Natalia Jurczuk, Marcin Kondraciuk, Katarzyna Ptaszyńska-Kopczyńska, Andrzej Raczkowski, Anna Szpakowicz, Karol Adam Kamiński

**Affiliations:** 1Department of Population Medicine and Civilization Diseases Prevention, Medical University of Bialystok, 15-269 Bialystok, Poland; mchlabicz@op.pl (M.C.); jacek909@wp.pl (J.J.); m.paniczko@gmail.com (M.P.); sowa@umb.edu.pl (P.S.); malgorzata.szpakowicz@umb.edu.pl (M.S.); magda.lapinska@umb.edu.pl (M.Ł.); n.jurczuk@gmail.com (N.J.); marcin.kondraciuk@umb.edu.pl (M.K.); andrzej.raczkowski@umb.edu.pl (A.R.); 2Department of Invasive Cardiology, Teaching University Hospital of Bialystok, 15-276 Bialystok, Poland; 3Department of Cardiology, Teaching University Hospital of Bialystok, 15-276 Bialystok, Poland; kasia.ptaszynska@op.pl (K.P.-K.); akodzi@poczta.onet.pl (A.S.)

**Keywords:** left ventricular hypertrophy, electrocardiography, android-type obesity, cardiovascular risk, population studies

## Abstract

Background: Left ventricular hypertrophy (LVH) is an important risk factor for cardiovascular events. The electrocardiography (ECG) has poor sensitivity, but it is commonly used to detect LVH. Aim: To evaluate the diagnostic efficacy of known ECG indicators to recognize LVH in subgroups with different cardiovascular risk levels. Methods: 676 volunteers were included. Results: We found that 10.2% of the analyzed population had LVH based on echocardiography. Individuals with LVH were older, had a higher body mass index, higher systolic blood pressure, lower heart rate, higher parameters of insulin resistance, higher cardiovascular risk, and android-type obesity. Variables that remained independently associated with LVH were QRS duration, left atrial volume index, troponin T, and hemoglobin A1c. The receiver operating characteristics (ROC) curve analysis of the Sokolow–Lyon index did not show a significant predictive ability to diagnose LVH in the whole study population including all cardiovascular risk classes. The ROC curves analysis of Cornell and Lewis indices showed a modest predictive ability to diagnose LVH in the general population and in a low cardiovascular class. Conclusions: There is a need for new, simple methods to diagnose LVH in the general population in order to properly evaluate cardiovascular risk and introduce optimal medical treatment of concomitant disease.

## 1. Introduction

Cardiovascular diseases (CVDs) are the most common diseases in the population but may be asymptomatic or misdiagnosed. Left ventricular hypertrophy (LVH) defined as an increased left ventricular mass (LVM), is a response to chronic pressure or volume overload and is an important risk factor for atrial fibrillation (AF), diastolic and systolic heart failure (HF), and a sudden death in patients with arterial hypertension (HA) [1,2,3,4,5,6,7]. The patients with LVH have a two-to-four-fold increased risk of cardiovascular (CV) morbidity and mortality compared to individuals with normal LVM [8]. Although echocardiography (ECHO) is a more sensitive tool for identifying LVH, it is cost-intensive and not always available. Therefore, an electrocardiogram (ECG) is suggested in the initial evaluation of individuals to detect LVH in the clinical setting [9]. Despite the fact that mortality from CVD remains unacceptably high [10], the incidence of CV risk factors is increasing, especially in developing countries [11]. The prevalence of LVH depending on the CV risk is not well established. The identification of individuals in the community with asymptomatic LVH may allow earlier diagnosis and initiation of treatment. It has to be stressed that the current understanding of CV risk management requires an analysis of particular factors in the context of the total risk. The total cholesterol concentration as well as blood pressure values are always analyzed together with the systematic coronary risk estimation (SCORE) of a particular patient [9]. Therefore, here we analyze the presence of LVH and diagnostic accuracy in the context of CV risk category, including the SCORE system. Given the increasing obesity epidemic in the general population, it appears that some ECG indices, especially based on the precordial leads, may have lower sensitivity than expected, hence their applicability must be verified in new studies.

**Aim:** To estimate the incidence of LVH in the general population; search for parameters associated with LVH; and evaluate diagnostic accuracy of the known electrocardiographic indices detecting LVH in the context of cardiovascular risk.

## 2. Methods

### 2.1. Study Population

The study was conducted in 2017–2019 on a representative sample of area residents. We examined 717 volunteers, randomly chosen from the local population aged between 20 to 79. Due to the lack of complete data, QRS complex duration ≥ 120 ms, fascicular blocks, bundle branch blocks, and paced rhythm, 41 people were excluded from further analysis. As a result, 676 people (mean age 48.65 ± 15.26 years, 59% female) were included in the study group. Sixty (8.9%) people had one or more established CVDs, specifically, coronary heart disease (CHD) 30 (4.4%), HF 8 (1.2%), peripheral artery disease (PAD) 6 (0.9%), stroke 6 (0.9%), heart surgery 2 (0.3%), or valvular heart defect 10 (1.5%). Moreover, 287 (42.5%) probands had a carotid atherosclerotic plaque on ultrasound examination, 192 (28.4%) a history of HA, 47 (7.0%) history of diabetes, 22 (3.3%) history of AF, and 141 (20.9%) were current smokers.

### 2.2. Data Collection and Assays

The details of the subjects’ medical history were collected from questionnaires at the time of study entry and included demographic data, CV risk factors, and a history of cardiovascular events. Peripheral intravenous fasting blood samples were collected at the time of visit, which always took place in the morning. Comprehensive assessments were performed. Anthropometric measurements including height, weight, and circumferences of waist, abdomen, hips, and thighs were taken. Body mass index (BMI) was calculated as weight in kilograms divided by height in meters squared. Waist-to-hip ratio (WHR) was calculated as a ratio between waist and hips circumference [12]. Blood pressure (BP) was measured using the oscillometeric method (obtained with the Omron Healthcare Co. Ltd MG Comfort device) after the participants were seated for at least 5 minutes. Resting ECG was performed using the AMEDTEC ECGpro CardioPart 12 USB (AMEDTEC Medizintechnik Aue GmbH, Aue, Germany). LVH in the ECG was defined using three different formulas: as >35 mm for men and women using the Sokolow–Lyon index (SV_1_ + RV_5_ or V_6_) [13]; >28 mm for men, >20 mm for women using the Cornell index (RaVL + SV_3_) [14]; >17 mm for both sexes using the Lewis index ((RI + SIII) − (RIII + SI)) [14]. The ECHO measurements were made using the B-mode ultrasound Vivid 9 (GE Healthcare, Chicago, IL, USA). In ECHO, measurements of the dimensions of the heart, left atrial (LA) volume, left ventricular ejection fraction (LVEF) using the Biplane method were performed. The left ventricular mass (LVM) was calculated using the Devereux Formula [15] LVM = 0.8 (1.04(IVST + LVID + LPWT)^3^ − (LVID)^3^ + 0.6), where IVST is interventricular septal thickness, LVID is  left ventricular internal dimension, and LPWT is left ventricle posterior wall thickness. Body surface area (BSA) was calculated by the formula: BSA = (W − 60) × 0.01 + H, where BSA is the body surface area in m^2^, W is the weight in kilograms, and H is height in meters [16]. The left ventricular mass index (LVMI) was calculated by the formula LVM/BSA (LVMI_BSA_), then the LVH was defined as LVMI ≥ 115 g/m^2^ for men and ≥95 g/m^2^ for women or LVMI was calculated by the formula LVM/height in m^2.7^ (LVMI_Height_) with LVH defined as LVMI ≥ 47g/m^2.7^ for women and LVMI ≥ 50 g/m^2.7^ for men (analyses in Appendix A) [17].

The left atrial volume index (LAVI) was calculated with enlarged volume defined as > 34 ml/m^2^. Diastolic dysfunction of the left ventricle was assessed based on the latest recommendations [18]. A carotid ultrasound was performed using two-dimensional ultrasound (Vivid 9, GE Healthcare, Chicago, IL, USA). The left and right common carotid arteries, carotid bifurcations, and internal and external carotid arteries were examined for wall thickness and the presence of atherosclerotic plaques. A plaque was defined as an area of focal wall thickening of at least 0.5 mm or 50% greater than the bsurrounding intima-media thickness (IMT) or IMT greater than 1.5 mm [19]. Stenosis severity was determined using the North American Symptomatic Carotid Endarterectomy Trial (NASCET) criteria [20]. Body composition was measured by the dual energy x-ray absorptiometry (DEXA) (GE Healthcare, Chicago, Ilinois, USA) with total body mass divided into three compartments: bone, fat mass, and lean mass. Fat mass index (FMI) was calculated as fat in kilograms divided by height in meters squared. The gynoid (G) and android (A) fat were measured automatically. The ranges for the android region of interest (ROI) were lower boundary at the pelvis cut, upper boundary above the pelvis cut by 20% of the distance between the pelvis and neck cuts, and lateral boundaries were the arm cuts. The ranges for gynoid ROI were upper boundary below the pelvis cut line by 1.5 times the height of the android ROI, gynoid ROI height equal to 2 times the height of the android ROI, and lateral boundaries were the outer leg cuts. The A/G ratio was calculated between the fat of the android (central) and fat of the gynoid (hip and thigh) regions. The gynoid/total fat mass (G/TF) ratio was calculated as the ratio between the gynoid fat and total fat. The android/TF (A/TF) ratio was calculated as the ratio between the android fat and total fat. The legs/TF ratio was calculated as the ratio between the leg fat and total fat. The study population was divided into CV risk classes according to the latest recommendation [21]. The Systematic Coronary Risk Estimation (SCORE) system was recalibrated in Poland [22]. Thus, we used the Pol-SCORE system to assess the 10-year risk of fatal CV disease based on the following risk factors: age, gender, smoking, BP, and total cholesterol [22,23]. To assess health related quality of life, the self-reported EQ-VAS (Euro Quality of Life Visual Analogue Scale) was utilized [24].

### 2.3. Ethical Issues

The ethical approval for this study was provided by the Ethics Committee of the Medical University of Bialystok (Poland) on 31 March 2016 (approval number: R-I-002/108/2016). The study was conducted in accordance with the Declaration of Helsinki and all participants gave written informed consent.

### 2.4. Statistical Analysis

Descriptive statistics for quantitative variables were presented as means and standard deviations, and as counts and frequencies for qualitative variables. Comparisons of continuous variables between subgroups were conducted using the Mann–Whitney or Kruskall–Wallis tests. The comparisons of categorical variables between subgroups were conducted using the Pearson’s chi-squared test. Associations between LVH using ECHO and other clinical and biochemical variables were analyzed using a multiple logistic regression model. Odds ratio (OR) was presented for unstandardized and standardized independent variables. Multiple regression (standardized for independent variables) and linear (standardized for independent and dependent variables) models were adjusted for demographics and variables associated with LVH: Model (1), age, sex, glomerular filtration rate (GFR), and BMI; Model (2), age, sex, GFR, BMI, history of HA, MI, IHD, PAD, DM, AF, stroke, and BP ≥1 40 and/or ≥ 90 mmHg; Model (3), age, sex, GFR: Model (4) age, sex, GFR, history of HA, MI, IHD, PAD, DM, AF, stroke, and BP ≥ 140 and/or ≥ 90 mmHg. The areas under the receiver operating characteristic curves (ROC) for all populations and for each of the CV risk groups were used to assess efficiency in recognizing LVH using the Sokolow–Lyon, Lewis and Cornell indices. The Yuden index was used to calculate sensitivity and specificity values of cut-off points for the ECG indices [25]. Statistical hypotheses were verified at a 0.05 significance level. IBM SPSS Statistics 20.0 statistical software (Armonk, NY, USA) was used for all calculations.

## 3. Results

The baseline characteristics of the study population are summarized in Table 1.

The mean age was 48.65 ± 15.26 years and 59% were female. We found 32 people (4.7%) meeting the ECG LVH criteria using the Sokolow–Lyon index, 17 (2.5%) using the Cornell index, and 23 (3.4%) using the Lewis index. On ECHO using LVMI_BSA_, LVH was diagnosed in 69 participants (10.2%). In a low-risk class, 5.1% of individuals had LVH, 10.6% had LVH in a moderate-risk class, and 17.7% had LVH in high-risk and very-high-risk classes (Table 2).

The comparison of groups with and without LVH using LVMI_BSA_ based on ECHO is presented in Table 3.

There was no gender difference (*p* = 0.944). Individuals with LVH were older (*p* < 0.001), had a higher BMI (*p* < 0.001), higher systolic blood pressure (BPs) (*p* < 0.001), lower heart rate (HR) (*p* = 0.044), and higher SCORE risk (*p* = 0.001). In addition, this group more often declared a history of HA, AF, DM, MI, CHD, and stroke and more often had an atherosclerotic plaque in a carotid artery.

In ECG, individuals with LVH had a longer P wave (*p* < 0.001) and QRS complex duration (*p* = 0.019), a higher Cornell index (*p* < 0.001) and Lewis index (*p* < 0.001), but not a Sokolow–Lyon index (*p* = 0.873). Lower heart rates can be explained by the fact that people with LVH were often on beta-blockers for a variety of indications. Both Chi2 test (Table 3) and the logistic regression analysis confirmed significant association between the use of beta-blockers and LVH (B = 1.119; *p* < 0.001). Patients taking beta-blockers had higher LVMI (85.7 ± 23.8 vs 77.1 ± 20.1 g/m^2^
*p* < 0.001).

In ECHO, there was no LVEF difference (*p* = 0.422), but LAVI (*p* < 0.001) was higher and diastolic dysfunction of LV was more common (*p* = 0.004) in individuals with LVH.

Participants with LVH had a higher concentration of N-terminal pro-brain natriuretic peptide (NT-proBNP), higher high-sensitivity troponin T (hs-TnT) concentration, higher C-reactive protein (CRP) levels and higher parameters of carbohydrate metabolism: fasting glucose, 120 min; glucose in an oral glucose tolerance test (OGTT), 120 min; insulin in OGTT hemoglobin A1c (HbA1c), a homeostatic model assessment of insulin resistance (HOMA-IR); and a lower glomerular filtration rate (GFR).

In the body composition analysis, LVH was associated with more fat tissue depicted by a higher FMI, higher body fat percentage (BFP), higher total fat (TF) mass, higher android fat mass, higher gynoid fat mass, higher legs fat mass, but not total lean mass, gynoid lean mass, or legs lean mass. Moreover, LVH was associated with a higher android/gynoid (A/G) fat mass ratio and android/total fat mass (A/TF) ratio, while with a lower gynoid/total fat mass (G/TF) ratio and legs/total fat (L/TF) mass ratio.

On the multivariable logistic regression analysis, variables that remained associated with LVH in Model 1 were QRS duration, LAVI, hs-TnT, HbA1c, the Cornel index, the Lewis index; in Model 2, the Sokolow–Lyon index appeared additionally (Table 4).

In Model 1 we used variables that have a proven relationship with LVH (age, sex, GFR, BMI), in Model 2 we added parameters that are associated with very high cardiovacsular risk (history of HA, MI, IHD, PAD, DM, AF, stroke, and BP ≥ 140 and/or ≥90 mmHg) to find parameters that are associated with LVH regardless of the CV risk class.

We removed BMI from Models 3 and 4 as a covariate in order to reveal the effect of fat tissue excess. The variables that remained associated with LVH in Model 3 were QRS duration, LAVI, hs-TnT, HbA1c, the Cornel index, and the Lewis index; in Model 4, the Sokolow–Lyon index remained irrelevant, and we confirmed the relationship between LVH and abdominal obesity regardless of occurrence of CV disease (Table 5).

In addition, we performed the multivariable linear regression analysis of LVMI_BSA_; variables that were positively correlated with LVMI_BSA_ in Models 1 and 2 were very similar to those found in logistic regression (Appendix A). In Model 3 after removing BMI as a covariate, we confirmed the relationship between LVMI and abdominal obesity (Appendix A).

In our study, 36.1% of the population was overweight and 23.8% obese. It is known that LVM indexed to BSA underestimates LVH prevalence in obese as well as overweight individuals, therefore, we calculated LVMI by the formula LVM/Height in m^2.7^. In Appendix A we showed the multivariable logistic regression analysis; variables that remained associated with LVH using LVMI_Height_ in Models 1 and 2 were QRS duration, LAVI, HbA1c, the Cornel index, and the Lewis index. In Models 3 and 4, hs-TnT, fasting glucose, 120 min glucose, fasting insulin, 120 min insulin, and indicators of abdominal obesity were associated with LVH. Moreover, in the multivariable linear regression analysis of LVMI_Height_, the relationship with carbohydrate metabolism and abdominal obesity were even more pronounced (Appendix A).

The ROC curve analysis of the Sokolow–Lyon index did not show a significant predictive ability to diagnose LVH in ECHO using LVMI_BSA_ in all study populations (AUC: 0.509; *p* = 0.816), as in all CV risk classes. At the same time, the ROC curve analysis of the Cornell index and Lewis index showed a modest, but statistically significant, predictive ability to diagnose LVH using LVMI_BSA_ in the general population (AUC: 0.658, *p* < 0.001; AUC: 0.687; *p* < 0.001, respectively) and in a low CV risk class (AUC: 0.671, *p* = 0.003; AUC: 0.797, *p* < 0.001, respectively). In high and very-high risk classes only, the Cornell index shows a weak predictive ability to diagnose LVH using LVMI_BSA_ (AUC: 0.659; *p* = 0.004) (Figure 1, Table 6).

Direct comparison of the AUCs from different ECG indices showed a significantly worse performance of the Sokolow–Lyon index in comparison to both Cornell and Lewis indices in all study populations and in a low CV risk class in comparison to the Lewis index (Table 6).

Table 7 shows the diagnostic accuracy of ECG criteria to detect LVH (using LVM/BSA index) in the study population according to CV classes at different sensitivities (90% or 80%) and specificities 90% or 80%).

The results of the EQ-VAS were analyzed. There were no significant differences between the individuals with and without LVH in ECHO using LVMI_BSA_ in each of the CV risk classes. This suggests that individuals with LVH may not be aware of their impaired health status. However, individuals in the low-risk group with LVH more often felt short of breath on exertion (Table 8) and more often sought medical help (Table 9) compared to individuals without LVH.

## 4. Discussion

The present study provides evidence on the frequency of LVH in the general population and on the ability of detection of LVH using known electrocardiographic indicators in terms of a cardiovascular risk level. The three main findings are (1) the higher than expected frequency of LVH in the low CV risk group (5.1%); (2) in the general population as well as in all CV risk classes, the ECG Sokolow–Lyon criteria for determining increased LVH are unreliable; and (3) while the Cornel voltage and Lewis voltage criteria are useful, they apply only to individuals with low CV risk. The diagnosis of LVH in the initially low risk patients would change their CV risk and may influence their treatment. Patients from the low CV risk group who had LVH on echocardiography more often complained of dyspnea and sought medical contact.

### 4.1. Clinical and Biochemical Factors

LVH is an abnormal increase in left ventricular mass (LVM), which is an important risk factor for HF, coronary events, stroke, PAD, arrhythmias, and mortality in patients with HA [1,2,3,4,5,6,7]. The detection of individuals with LVH may facilitate further diagnostics and treatment, which is extremely important because regression of LVH is associated with lower incidence of CV events and improved cardiac function [6,26,27,28,29]. Patients with risk factors that are controlled have a lower probability of developing complications than patients whose CV risk factors are unrecognized or poorly controlled [30].

Therefore, early diagnosis of LVH is essential. In the current study, 10.2% of individuals were found to have LVH, and there was no gender difference. In earlier studies, the prevalence of LVH in the total population was 14.9% for men and 9.1% for women (LVH based ECHO and defined as LVMI ≥ 145.5 g/m^2^ for men and ≥ 125.5 g/m^2^ for women) [30], in another study, 10% of obese individuals had LVH (based magnetic resonance imaging (MRI), LVH was defined as LVMI > 51.9 g/m^1.7^ in men and >41.8 g/m^1.7^ in women) [31]. Tanaka [32] revealed, that prevalence of LVH in non-hypertensive individuals using the ECG criteria of the Sokolow–Lyon index was 11.7%, and using the Cornell voltage criteria, 1.9%. Moreover, the present study reported that 5.1% of individuals had LVH in the low-risk class and 10.6% in the moderate-risk class. Schillaci [33] showed in never-treated hypertensive patients that the prevalence of LVH was 21% and 32% in low-risk and medium-risk groups, respectively. Mancusi showed that 37% of low-risk hypertensive patients had LVH in ECHO (LVH defined as >47.0 g/m^2.7^ in women and >50.0 g/m^2.7^ in men) [34]. Karakan [35] showed that the independent factors affecting LVMI in a group of hypertensive patients were age, weight, gender, and body fat percentage (BFP). Similarly, in the current study in the general population, individuals with LVH were older and had higher BPs, BMI, and BFP. However, there were no gender differences, and additionally individuals with LVH had higher SCORE and lower HR.

The development of LVH is highly correlated with systolic hypertension [36] mediated by mechanical stress of pressure overload and also by various neurohormonal substances that exert trophic effects on myocytes [37], including angiotensin II, aldosterone, norepinephrine, and insulin, and promote matrix deposition and myocyte hypertrophy independently of effects on systemic arterial pressure [38]. These substances stimulate the production of cytokines and growth factors (transforming growth factor beta, fibroblast growth factor, insulin growth factor, and others). However, LVM is also dependent on genetic factors, obesity, insulin resistance, and alcohol intake [39,40,41].

Android (central) obesity is associated with an increased risk for metabolic dysfunction compared to gynoid (femoral/gluteal) obesity [42,43]. Individuals with android-type obesity compared to gynoid-type obesity have higher fasting insulin and glucose levels [44,45]. Metabolic syndrome, even after controlling for age, sex, and BPs, has been associated with increased LVM [46]. In our study, individuals with LVH had a higher level of fasting glucose, glucose after 120 min in OGTT, insulin after 120 min in OGTT, HOMA-IR, and HbA1c; these in combination with android-type obesity can confirm earlier reports of an association of insulin resistance with myocyte trophic effects [37,46]. This phenomenon was even more pronounced in the analysis using LVM indexed to height in m^2.7^, which is less dependent on body weight.

We have noticed a correlation between a lower HR and LVH. Inoune [47] showed a similar dependence, in which an elevated resting HR was negatively associated with the development of ECG LVH in a healthy male group. We explain this correlation by more common use of beta-blockers among individuals with LVH, who more frequently had a history of HA, myocardial infarction (MI), CHD, and AF.

The QRS duration, hs-TnT, HbA1c, and LAVI were also associated with LVH in ECHO using both methods, LVMI_BSA_ or LVMI_Height_. In another study, QRS duration was similarly described as an independent predictor of LVH in the absence of aberrant conduction or a bundle branch block [48]. The hs-TnT is an intrinsic myocyte protein, with myocyte turnover associated with its level, reflecting pathophysiological mechanisms, including cardiac cell death, apoptosis, and probably cardiomyocyte strain, and it correlates with CV risk factors, cardiac hypertrophy, and metabolic disorders [49,50]. HbA1c was found to be a predictor of coronary artery disease, left ventricular dysfunction [51,52], and LVH [32]. We confirmed the correlation of HbA1c with LVH in the current study. Left atrial enlargement (LAE) is closely related to LVH and to diastolic dysfunction [53]. We did not find the NT-proBNP association with LVH. Other papers have reported that brain natriuretic peptide (BNP) is suboptimal for LVH screening [54,55]. This could be explained by a positive association of NT-proBNP concentration with gynoid fat mass and G/TF mass index, and a negative correlation with A/G fat mass index in the general population [56]. Furthermore, another variable could be the study population, which included mainly healthy members of the public and the patients with ECG abnormalities typically associated with heart failure, like bundle branch block, were excluded from the analysis. In individuals with LVH, android-type obesity is predominant, which may affect the relative reduction of NT-proBNP concentration in this population [56].

### 4.2. Electrocardiographic Indicators

Although ECHO is a more sensitive tool for identifying LVH, it is not always available, moreover, it is cost-intensive, time-consuming, and expert-dependent. Therefore, ECG is suggested in the initial evaluation of individuals to detect LVH [9], which may allow for early initiation of further diagnostics, therapy implementation, and a delay or prevention of progression to heart failure [57,58]. The conditions to screen for in pre-clinical stages should be of public health importance. The screening method should be safe and the condition should have an effective treatment. Moreover, high diagnostic sensitivity is in particular demand in population screening. ECG is affordable, simple to perform, and widely used, but its poor diagnostic accuracy and low sensitivity of ECG criteria limit its use in detecting LVH [59,60]. Pewsner [61] showed that in hypertensive patients in a systematic review, the sensitivity of the Sokolow–Lyon voltage criteria ranged from 8% to 40% at specificities of 53–100%, and the sensitivity of the Cornell voltage criteria ranged from 2% to 19% at specificities of 89–100%. We confirmed in the current analysis the low sensitivity of the Sokolow–Lyon voltage criteria, as all study population were 11.59% or 18.84% at specificities of 90% or 80%, respectively. The sensitivity of the Cornell voltage criteria was better for all study population with 21.74% or 36.23% at specificities of 90% or 80%, respectively. In previous studies, the proposed criteria for ECG diagnosis of LVH had typically high (>90%) specificities with lower sensitivities (6.9–60%) [59], rendering the ECG a relatively ineffective method for identification of individuals with LVH. Maybe we should consider increasing sensitivity at the expense of specificity or adding new non-ECG parameters that may increase sensitivity without compromising specificity.

We have shown that the Sokolow–Lyon voltage criteria for recognition of LVH were the weakest, which is in line with other studies. Elffers et al. [31] showed that discriminative performance of the Sokolow–Lyon voltage criteria is poor (AUC: 0.58) in obese individuals, while Ricciardi et al. [62] showed that the Cornel voltage and Cornel product criteria have a more accurate diagnostic performance compared with other hospitalized patients. Courad et al. [63] suggested that the R wave in a VL lead should be the first index used to detect LVH of hypertensive patients, and it should replace the Sokolow–Lyon index due to the latter’s inadequacies. On the other hand, in the Losartan Intervention For Endpoint (LIFE) study, obese and overweight hypertensive patients had a lower Sokolow–Lyon index in comparison to normal-weight individuals, and a lower prevalence of recognition LVH using a lower Sokolow–Lyon voltage [64]. Elffers [31] suggested that the precordial leads voltage affected the Sokolow–Lyon voltage more than the Cornell voltage. The reason could be the increase of epicardial fat resulting in a higher distance between the heart and the skin electrodes on the chest wall. Thus, decreasing sensitivity of the Sokolow–Lyon index in the population with visceral obesity [31,65].

### 4.3. Health Related Quality of Life

The EQ-5D was designed to measure health related quality of life. This instrument is widely used in the health sector in patient-reported outcome exercises, population heath studies, and health technology assessments [24,66]. In our study, we applied it to investigate if patients who have LVH, and thus are at higher CV risk, may have symptoms that could cause them to seek medical advice. There were no significant differences between the individuals with LVH and other study participants. This suggests that the individuals with LVH may not be aware of their impaired health status and increased CV risk. However, individuals in the low-risk group with LVH more often felt dyspnea on exertion and more often sought medical help when compared to individuals without LVH.

### 4.4. Limitation

The small number of subjects with LVH per group could potentially affect the reliability of the results. We treated subgroup analyses in this study as hypothesis-generating, therefore, this part should be interpreted with caution. The left ventricular mass and left ventricular mass indices were estimated by using two-dimensional echocardiography, despite superior accuracy of cardiac magnetic resonance imaging. The research did not improve the analysis in patients with right or left bundle branch block, incomplete bundle branch block, paced rhythm, or QRS duration longer than 120 ms. These subgroups were excluded from the study.

### 4.5. Clinical Applications

The present study gives an argument that currently used ECG criteria (especially the widely used Sokolow–Lyon index) for recognizing LVH are insufficient and their efficacy may differ depending on initial CV risk level. Patients with android type obesity, symptoms (unexplained dyspnea) and additional CV risk factors should be screened for LVH with ECHO even in the absence of ECG criteria for LVH.

## 5. Conclusions

About 10% of the general population has LVH and it is associated with android-type obesity, LAVI, QRS duration, hs-TnT, and HbA1c. There is a higher than expected frequency of LVH in the low CV risk group (5.1%). Both Cornell and Lewis indices are more efficient than Sokolow–Lyon in diagnosing LVH. Currently used ECG indices have a very poor predictive value and are applicable only in the low CV risk group. Consequently, there is an urgent need for new, simple methods to diagnose LVH in the general population.

We confirmed in many analyses a positive significant correlation between android-type obesity and LVMI or LVH (Table 5, Appendix A).

## Figures and Tables

**Figure 1 jcm-09-01364-f001:**
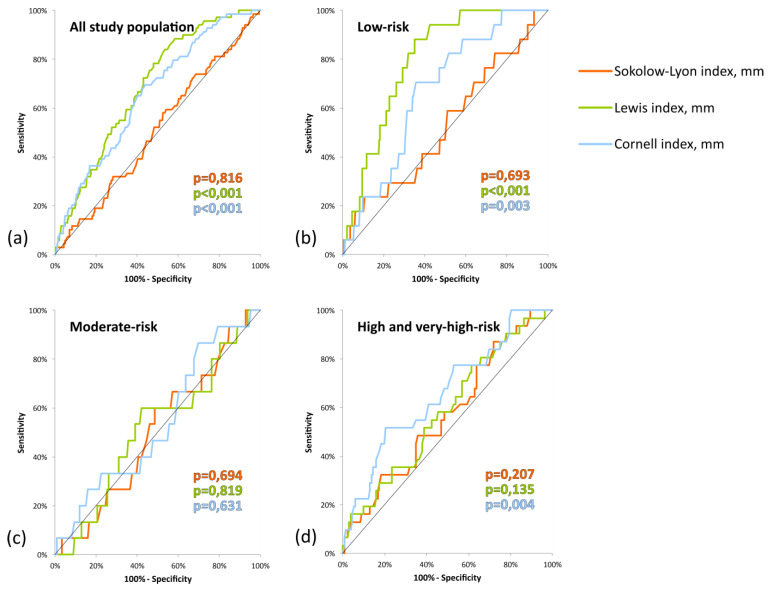
Receiver operating characteristic curves of the Sokolow–Lyon index, Lewis index, and Cornell index for recognized left ventricular hypertrophy in echocardiography (ECHO) using left ventricular mass indexed to body surface area (LVMI_BSA_): (**a**) all study population; (**b**) low cardiovascular risk; (**c**) moderate cardiovascular risk; (**d**) high and very-high cardiovascular risk.

**Table 1 jcm-09-01364-t001:** Characteristics of the study population.

Variable Value (*n* = 676)	
Age, years	48.65 ± 15.26
Male sex	277 (41.0)
SCORE Risk, %	4.31
BMI, kg/m^2^	26.70 ± 4.85
BMI < 25 kg/m^2^	271 (40.1)
BMI 25–29.99 kg/m^2^	244 (36.1)
BMI ≥ 30 kg/m^2^	161 (23.8)
NT-proBNP, pg/mL	90.02 ± 183.06
hs-TnT, pg/mL	6.40 ± 4.95
LVEF biplane, %	58.13 ± 5.68
Diastolic dysfunction of left ventricle	84 (12.4)
LVMI_BMI_ *, g/m^2^	78.64 ± 21.04
LVMI_BMI_ *, ≥95 g/m^2^ women, ≥115 g/m^2^ men	69 (10.2)
LAVI, mL/m^2^	22.60 ± 7.02
LAVI, >34 mL/m^2^	40 (5.9)
Sokolow–Lyon index, mm	22.04 ± 7.09
Sokolow–Lyon index, >35 mm	32 (4.7)
Cornell index, mm	11.71 ± 5.36
Cornell index, >28 mm men, >20 mm women	17 (2.5)
Lewis index, mm	2.14 ± 8.40
Lewis index, >17 mm	23 (3.4)
Creatinine, μmoL/L	70.83 ± 15.34
GFR, mL/min/1.73 m^2^	113.84 ± 35.74
BPs, mmHg	123.53 ± 18.07
BPd, mmHg	81.75 ± 10.39
BP ≥ 140 and/or ≥90 mmHg	167 (24.7)
HR, bpm	72.37 ± 11.09
Carotid plaque	287 (42.5)
History of hypertension	192(28.4)
History of diabetes	47 (7.0)
History of atrial fibrillation	22 (3.3)
History of myocardial infarction	14 (2.1)
History of coronary heart disease	16 (2.4)
History of heart failure	8 (1.2)
History of heart defect	10 (1.5)
History of peripheral artery disease	6 (0.9)
History of stroke	6 (0.9)
Currently smoking	141 (20.9)

The data are shown as *n* (%), mean ± SD. SD: standard deviation; SCORE: Systematic Coronary Risk Estimation; BMI: body mass index; kg: kilogram; m^2^: square meter; NT-proBNP: N-terminal pro-brain natriuretic peptide; hs-TnT: high-sensitivity troponin T; LVEF Biplane: left ventricular ejection fraction biplane Simpson’s method; LVMI: left ventricular mass index; LAVI: left atrial volume index; GFR: glomerular filtration rate Cockcroft–Gault Equation; BPs: systolic blood pressure; BPd: diastolic blood pressure; mmHg: millimeters of mercury; HR: heart rate; bpm: beats per minute. * The left ventricular mass (LVM) index was calculated by the formula LVM/body surface area (BSA).

**Table 2 jcm-09-01364-t002:** Characteristics of the population according to cardiovascular risk.

Variable	Cardiovascular Risk			
	Low Risk *n* = 345	Moderate Risk *n* = 147	High Risk *n* = 108	Very High Risk *n* = 67
Male sex	130 (37.7)	58 (39.5)	45 (41.7)	38 (56.7)
Age, years	39.33 ± 13.14	55.17 ± 7.96	61.75 ± 9.32	65.79 ± 7.65
LVMI *, g/m^2^	73.25 ± 19.18	80.70 ± 18.34	85.37 ± 21.01	89.46 ± 24.37
LVMI *, ≥95 g/m^2^ women, ≥115 g/m^2^ men	17 (5.1)	15 (10.6)	20 (19.8)	11 (17.7)
Use of beta-blockers	21 (6.1)	38 (25.9)	30 (27.8)	37 (55.2)
Sokolow–Lyon index, mm	22.67 ± 7.20	21.02 ± 6.63	20.82 ± 6.73	20.62 ± 6.43
Sokolow–Lyon index, >35 mm	22 (6.4)	3 (2.0)	3 (2.8)	1 (1.5)
Cornell index, mm	10.78 ± 5.49	11.74 ± 4.26	12.71 ± 5.09	14.43 ± 5.58
Cornell index, >28 mm men, >20 mm women	9 (2.6)	2 (1.4)	1 (0.9)	3 (4.5)
Lewis index, mm	−0.24 ± 8.47	4.54 ± 7.01	5.75 ± 7.59	5.50 ± 7.05
Lewis index, >17 mm	7 (2.0)	7 (4.8)	6 (5.6)	4 (6.0)

The data are shown as *n* (%), mean ± SD. SD, standard deviation; LVMI, left ventricular mass index; * The left ventricular mass (LVM) index was calculated by the formula LVM/BSA.

**Table 3 jcm-09-01364-t003:** Clinical, biochemical, electrocardiographic, echocardiographic, and body composition characteristics compared between individuals with and without left ventricular hypertrophy using echocardiography (ECHO) in the general population.

Variable	Study Population (*n* = 676)		
	Subjects without LVH_BSA_ *	Subjects with LVH_BSA_ *	*p*-Values
Sample size	607 (89.8)	69 (10.2)	
Male sex	249 (41.0)	28 (40.6)	0.944
Age, years	47.37 ± 15.08	59.93 ± 11.91	<0.001
BMI, kg/m^2^	26.34 ± 4.74	29.87 ± 4.71	<0.001
BMI < 25 kg/m^2^	261 (43.3)	10 (14.5)	<0.001
BMI 25–29.99 kg/m^2^	218 (35.9)	26 (37.7)	<0.001
BMI ≥ 30 kg/m^2^	128 (21.1)	33 (47.0)	<0.001
SCORE Risk, %	3.35	4.47	0.001
Carotid plaque	238 (39.2)	49 (71.0)	<0.001
BPs, mmHg	122.65 ± 17.94	131.46 ± 17.44	<0.001
BPd, mmHg	81.57 ± 10.34	83.38 ± 10.72	0.149
HR, bpm	72.65 ± 11.11	69.80 ± 10.65	0.044
LVEF BP, %	58.22 ± 5.57	57.28 ± 6.56	0.422
LVMI_BSA_, g/m^2^	74.26 ± 16.20	117.20 ± 19.34	<0.001
LAVI, ml/m^2^	22.16 ± 6.44	26.49 ± 10.12	<0.001
Diastolic dysfunction of left ventricle	68 (11.2)	16 (23.2)	0.004
P wave time, ms	105.82 ± 10.17	112.09 ± 14.48	<0.001
QRS time, ms	90.59 ± 9.42	93.07 ± 8.48	0.019
Sokolow–Lyon index, mm	22.00 ± 7.04	22.38 ± 7.52	0.818
Sokolow–Lyon index, > 35 mm	29 (4.8)	3(4.3)	0.873
Cornell index, mm	11.40 ± 5.24	14.49 ± 5.69	<0.001
Cornell index, >28 mm men, >20 mm women	14 (2.3)	3 (4.3)	0.305
Lewis index, mm	1.57 ± 8.30	7.08 ± 7.65	<0.001
Lewis index, >17 mm	16 (2.6)	7 (10.1)	0.001
NT-proBNP, pg/mL	72.79 ± 79.91	212.81 ± 460.96	<0.001
hs-TnT, pg/mL	5.94 ± 3.81	9.73 ± 9.17	<0.001
Fasting glucose, mg/dL	99.63 ± 14.73	111.22 ± 34.21	0.002
120 min glucose, mg/dL	121.46 ± 38.06	147.96 ± 42.18	<0.001
Fasting insulin, µUL/mL	11.32 ± 6.69	13.40 ± 8.90	0.076
120 min Insulin, µUL/mL	58.12 ± 51.52	89.27 ± 118.74	0.004
HbA1c, %	5.43 ± 0.48	5.90 ± 0.94	<0.001
HOMA-IR	2.84 ± 2.04	3.71 ± 2.87	0.013
GFR, mL/min/1.73 m^2^	114.76 ± 35.98	105.77 ± 32.75	0.045
hsCRP, mg/L	1.64 ± 3.36	2.67 ± 4.76	0.001
Waist, cm	85.68 ± 13.17	93.91 ± 11.17	<0.001
Hips, cm	98.73 ± 9.22	104.44 ± 9.44	<0.001
WHR	0.87 ± 0.10	0.90 ± 0.09	0.008
FMI, kg/m^2^	8.94 ± 3.50	11.16 ± 3.54	<0.001
% fat	33.18 ± 7.50	35.55 ± 7.06	0.017
Total fat mass, kg	25.38 ± 9.02	31.58 ± 8.74	<0.001
Total lean mass, kg	48.39 ± 10.70	50.08 ± 9.83	0.105
Android fat mass, kg	2.30 ± 1.20	3.02 ± 1.10	<0.001
Android lean mass, kg	3.30 ± 0.71	3.53 ± 0.68	0.009
Gynoid fat mass, kg	3.98 ± 1.37	4.59 ± 1.50	0.001
Legs fat mass, kg	7.58 ± 2.72	8.52 ± 3.11	0.021
Visceral mass, kg	1.12 ± 0.92	1.78 ± 0.98	<0.001
A/G fat mass ratio	0.57 ± 0.22	0.68 ± 0.22	<0.001
G/T fat mass ratio	0.16 ± 0.03	0.15 ± 0.02	0.005
A/T fat mass ratio	0.09 ± 0.02	0.10 ± 0.02	<0.001
Legs/T fat mass ratio	0.31 ± 0.07	0.28 ± 0.06	0.001
History of hypertension	159 (26.2)	33 (47.8)	<0.001
History of diabetes	34 (5.6)	13 (18.8)	<0.001
History of atrial fibrillation	15 (2.5)	7 (10.1)	0.001
History of myocardial infarction	10 (1.6)	4 (5.8)	0.022
History of coronary heart disease	12 (2.0)	4 (5.8)	0.049
History of heart failure	9 (1.5)	1 (1.4)	0.979
History of valvular heart defect	8 (1.3)	0 (0.0)	0.336
History of peripheral artery disease	4 (0.7)	2 (2.9)	0.061
History of stroke	3 (0.5)	3 (4.3)	0.001
Use of beta-blockers	97 (16.0)	26 (37.7)	<0.001
Currently smoking	125 (20.6)	16 (23.2)	0.630

The data are shown as *n* (%), mean ± SD. SD: standard deviation; LVH: left ventricular hypertrophy; BMI: body mass index; kg: kilogram; m^2^: square meter; SCORE: Systematic Coronary Risk Estimation; BPs: systolic blood pressure; BPd: diastolic blood pressure; mmHg: millimeters of mercury; HR: heart rate; bpm: beats per minute; LVEF BP: left ventricular ejection fraction biplane Simpson’s method; LVMI: left ventricular mass index; LAVI: left atrial volume index; NT-proBNP: N-terminal pro-brain natriuretic peptide; hs-TnT: high-sensitivity troponin T; HbA1c: hemoglobin A1c; HOMA-IR: homeostasis model assessment of insulin resistance; LDL: low-density lipoprotein, HDL: high-density lipoprotein; TG: triglycerides; GFR: glomerular filtration rate Cockcroft–Gault Equation, CRP: C-reactive protein; SHBG: sex hormone binding globulin; WHR: waist–hip ratio; FMI: fat mass index; A: android; G: gynoid; T: total. * The left ventricular mass (LVM) index was calculated by the formula LVM/BSA, and the LVH was defined as LVMI ≥ 115 g/m^2^ for men and ≥ 95 g/m^2^ for women.

**Table 4 jcm-09-01364-t004:** Results of the left ventricular hypertrophy * multivariable logistic regression analysis in the study population.

Variable	Model 1				Model 2			
	OR Unstandardized	95% C.I.	OR Standardized **	*p*	OR Unstandardized	95% C.I.	OR Standardized **	*p*
BPs, mmHg	1.010	0.994–1.026	1.194	0.216	-	-		-
BPd, mmHg	1.004	0.977–1.032	1.043	0.773	-	-		-
HR, bpm	0.986	0.963–1.011	0.859	0.271	0.983	0.959–1.009	0.829	0.192
LVEF BP, %	1.004	0.958–1.053	1.024	0.862	1.014	0.964–1.067	1.083	0.589
LAVI, ml/m^2^	1.044	1.008–1.082	1.356	0.016	1.045	1.007–1.086	1.366	0.021
P wave time, ms	1.021	0.994–1.049	1.253	0.129	1.017	0.989–1.045	1.199	0.236
QRS time, ms	1.046	1.010–1.083	1.519	0.012	1.044	1.007–1.082	1.496	0.019
Sokolow–Lyon index, mm	1.041	0.999–1.085	1.332	0.054	1.044	1.001–1.089	1.357	0.047
Cornell index, mm	1.080	1.026–1.137	1.510	0.003	1.071	1.015–1.129	1.442	0.012
Lewis index, mm	1.042	1.005–1.081	1.417	0.025	1.044	1.006–1.083	1.433	0.024
NT-proBNP, pg/mL	1.002	1.000–1.004	1.413	0.088	1.002	1.000–1.004	1.350	0.118
hs-TnT, pg/mL	1.069	1.005–1.137	1.392	0.034	1.077	1.006–1.152	1.441	0.032
Fasting glucose, mg/dL	1.007	0.996–1.019	1.142	0.218	1.008	0.994–1.022	1.152	0.284
120 min glucose, mg/dL	1.004	0.997–1.012	1.176	0.274	1.006	0.998–1.013	1.246	0.154
Fasting insulin, µUL/mL	0.983	0.942–1.026	0.888	0.434	0.998	0.955–1.043	0.989	0.943
120 Insulin, µUL/mL	1.001	0.997–1.005	1.062	0.623	1.002	0.998–1.006	1.156	0.252
HbA1c, %	1.755	1.149–2.680	1.372	0.009	1.940	1.101–3.418	1.451	0.022
HOMA-IR	0.966	0.849–1.100	0.928	0.604	1.000	0.872–1.147	1.000	0.998
hsCRP, mg/L	1.019	0.965–1.076	1.070	0.496	1.021	0.966–1.079	1.076	0.460
WHR	0.126	0.002–9.095	0.819	0.342	0.166	0.002–13.570	0.841	0.424
% fat	0.001	0.000–4.083	0.613	0.107	0.003	0.000–10.576	0.650	0.165
Total fat mass, kg	0.951	0.880–1.028	0.635	0.210	0.955	0.880–1.036	0.655	0.267
Total lean mass, kg	1.038	0.980–1.100	1.490	0.203	1.037	0.977–1.101	1.471	0.230
Android fat mass, kg	0.722	0.434–1.199	0.673	0.208	0.764	0.444–1.314	0.721	0.330
Gynoid fat mass, kg	1.005	0.690–1.465	1.008	0.977	1.037	0.700–1.535	1.051	0.858
Legs fat mass, kg	0.980	0.832–1.154	0.945	0.807	0.987	0.831–1.174	0.965	0.885
Visceral mass, kg	1.039	0.631–1.708	1.036	0.881	1.056	0.621–1.798	1.053	0.840
A/G fat mass ratio, %	0.999	0.980–1.018	0.970	0.890	1.001	0.981–1.021	1.015	0.949
G/T fat mass ratio, %	1.063	0.907–1.245	1.180	0.453	1.068	0.905–1.262	1.197	0.436
A/T fat mass ratio, %	1.034	0.829–1.290	1.075	0.766	1.079	0.856–1.361	1.180	0.518
Legs/T fat mass ratio, %	0.993	0.934–1.055	0.949	0.811	0.991	0.929–1.057	0.938	0.780
Risk SCORE, %	0.995	0.914–1.084	1.000	0.912	0.977	0.888–1.074	0.999	0.626

BPs: systolic blood pressure; BPd: diastolic blood pressure; mmHg: millimeters of mercury; HR: heart rate; bpm: beats per minute; LVEF BP: left ventricular ejection fraction biplane Simpson’s method; LAVI: left atrial volume index; NT-proBNP: N-terminal pro-brain natriuretic peptide; hs-TnT: high-sensitivity troponin T; HbA1c: hemoglobin A1c; HOMA-IR: homeostasis model assessment of insulin resistance; CRP: C-reactive protein; WHR: waist-hip ratio; A: android; G: gynoid, T: total; SCORE: Systematic Coronary Risk Estimation; GFR: glomerular filtration rate Cockcroft-Gault Equation; BMI: body mass index; Model 1: adjusted for age, sex, GFR, BMI; Model 2: model 1 + additional adjustment for: history of hypertension, diabetes, atrial fibrillation, myocardial infarction, coronary heart disease, heart failure, peripheral artery disease, stroke and BP ≥ 140 and/or ≥90 mmHg. * The left ventricular mass (LVM) index was calculated by the formula LVM/BSA, and the LVH was defined as LVMI ≥ 115 g/m^2^ for men and ≥95 g/m^2^ for women. ** Standardized for independent variables.

**Table 5 jcm-09-01364-t005:** Results of the left ventricular hypertrophy * multivariable logistic regression analysis in the study population.

Variable	Model 3				Model 4			
	OR Unstandardized	95% C.I.	OR Standardized **	*p*	OR Unstandardized	95% C.I.	OR Standardized **	*p*
BPs, mmHg	1.012	0.997–1.027	1.242	0.120	-	-	-	-
BPd, mmHg	1.010	0.983–1.038	1.112	0.461	-	-	-	-
HR, bpm	0.988	0.964–1.012	0.871	0.316	0.984	0.960–1.010	0.840	0.224
LVEF BP, %	1.001	0.955–1.050	1.008	0.956	1.011	0.961–1.063	1.063	0.675
LAVI, ml/m^2^	1.047	1.011–1.084	1.381	0.009	1.048	1.010–1.088	1.389	0.014
P wave time, ms	1.027	1.001–1.055	1.341	0.042	1.022	0.995–1.050	1.261	0.118
QRS time, ms	1.048	1.014–1.084	1.553	0.006	1.046	1.010–1.082	1.517	0.012
Sokolow–Lyon index, mm	1.027	0.988–1.068	1.211	0.176	1.034	0.992–1.077	1.264	0.112
Cornell index, mm	1.082	1.029–1.137	1.525	0.002	1.071	1.017–1.128	1.443	0.010
Lewis index, mm	1.053	1.017–1.090	1.540	0.004	1.053	1.016–1.091	1.539	0.005
NT-proBNP, pg/mL	1.002	1.000–1.005	1.537	0.058	1.002	1.000–1.004	1.468	0.077
hs-TnT, pg/mL	1.078	1.014–1.147	1.453	0.016	1.081	1.012–1.156	1.472	0.021
Fasting glucose, mg/dL	1.011	1.000–1.023	1.224	0.056	1.011	0.997–1.025	1.218	0.132
120 min glucose, mg/dL	1.007	1.000–1.014	1.300	0.063	1.008	1.000–1.015	1.344	0.046
Fasting insulin, µUL/mL	1.011	0.975–1.048	1.079	0.555	1.019	0.981–1.060	1.145	0.329
120 Insulin, µUL/mL	1.002	0.999–1.006	1.154	0.231	1.003	0.999–1.008	1.236	0.103
HbA1c, %	1.961	1.299–2.961	1.460	0.001	2.127	1.212–3.731	1.528	0.009
HOMA -IR	1.045	0.936–1.167	1.100	0.434	1.063	0.942–1.199	1.142	0.322
hsCRP, mg/L	1.029	0.975–1.085	1.106	0.296	1.028	0.975–1.084	1.103	0.306
WHR	3.194	0.074–137.489	1.118	0.545	2.485	0.048–127.726	1.092	0.651
% fat	21.797	0.052–9204.292	1.261	0.318	16.272	0.031–8603.827	1.234	0.383
Total fat mass, kg	1.050	1.011–1.090	1.558	0.011	1.046	1.006–1.088	1.510	0.024
Total lean mass, kg	1.076	1.025–1.130	2.184	0.003	1.071	1.019–1.125	2.067	0.007
Android fat mass, kg	1.390	1.058–1.826	1.491	0.018	1.380	1.027–1.853	1.478	0.032
Gynoid fat mass, kg	1.406	1.104–1.793	1.609	0.006	1.384	1.082–1.771	1.573	0.010
Legs fat mass, kg	1.147	1.018–1.293	1.462	0.025	1.143	1.012–1.292	1.450	0.032
Visceral mass, kg	1.608	1.124–2.300	1.567	0.009	1.581	1.064–2.347	1.542	0.023
A/G fat mass ratio, %	1.013	0.997–1.030	1.350	0.112	1.013	0.995–1.031	1.340	0.151
G/T fat mass ratio, %	0.978	0.844–1.133	0.941	0.765	1.001	0.857–1.171	1.004	0.985
A/T fat mass ratio, %	1.232	1.028–1.476	1.572	0.024	1.246	1.026–1.512	1.610	0.026
Legs/T fat mass ratio, %	0.959	0.907–1.014	0.746	0.137	0.964	0.909–1.023	0.776	0.226
Risk SCORE, %	0.988	0.909–1.075	0.999	0.788	0.969	0.883–1.063	0.998	0.504

BPs: systolic blood pressure; BPd: diastolic blood pressure; mmHg: millimeters of mercury; HR: heart rate; bpm: beats per minute; LVEF BP: left ventricular ejection fraction biplane Simpson’s method; LAVI: left atrial volume index; NT-proBNP: N-terminal pro-brain natriuretic peptide; hs-TnT: high-sensitivity troponin T; HbA1c: hemoglobin A1c; HOMA-IR: homeostasis model assessment of insulin resistance; CRP: C-reactive protein; WHR: waist-hip ratio; A: android; G: gynoid, T: total; SCORE: Systematic Coronary Risk Estimation; GFR: glomerular filtration rate Cockcroft-Gault Equation; Model 3: adjusted for age, sex, GFR; Model 4: model 3 + additional adjustment for: history of hypertension, diabetes, atrial fibrillation, myocardial infarction, coronary heart disease, heart failure, peripheral artery disease, stroke and BP ≥ 140 and/or ≥ 90 mmHg. * The left ventricular mass (LVM) index was calculated by the formula LVM/BSA, and the LVH was defined as LVMI ≥ 115 g/m2 for men and ≥ 95 g/m2 for women. ** Standardized for independent variables.

**Table 6 jcm-09-01364-t006:** Diagnostic accuracy of electrocardiographic criteria to detect the left ventricular hypertrophy * in study population and according cardiovascular risk classes.

Study Population	Variable							
	Cut-Off Point, mm	Sensitivity, %	Specificity, %	NPV, %	PPV, %	AUC	95% C.I. (AUC)	*p*
**All Study Population**								
Sokolow–Lyon index, mm	>20.75	57.97	47.45	90.85	11.14	0.5085	0.437–0.580	0.8162
Cornell index, mm	>11.55	69.57	56.01	94.18	15.24	0.6575 **	0.593–0.722	<0.001
Lewis index, mm	>0.15	84.06	46.62	96.26	15.18	0.6869 **	0.628–0.745	<0.001
**Low-Risk**								
Sokolow–Lyon index, mm	<15.05	23.53	89.34	95.64	10.53	0.5300	0.381–0.679	0.6932
Cornell index, mm	>11.55	70.59	64.26	97.62	9.52	0.6708	0.559–0.783	0.0027
Lewis index, mm	>0.75	88.24	64.89	99.04	11.81	0.7966 **	0.718–0.875	<0.001
**Moderate-Risk**								
Sokolow–Lyon index, mm	>21.05	60.00	51.59	91.55	12.86	0.5034	0.355–0.652	0.9638
Cornell index, mm	<13.75	86.67	30.16	95.00	12.87	0.5376	0.384–0.691	0.6309
Lewis index, mm	<3.35	60.00	57.94	92.41	14.52	0.5183	0.362–0.674	0.8187
**High and Very-High Risk**								
Sokolow–Lyon index, mm	>16.20	87.10	28.03	90.24	22.13	0.5714	0.461–0.682	0.2066
Cornell index, mm	>16.30	51.61	79.55	87.50	37.21	0.6581	0.550–0.766	0.0042
Lewis index, mm	>2.50	77.42	38.64	87.93	22.86	0.5844	0.474–0.695	0.1351

AUC, area under the receiver operating characteristic curve; C.I.: confidence interval; NPV: negative predictive value; PPV: positive predictive value. * The left ventricular mass (LVM) index was calculated by the formula LVM/BSA, and the LVH was defined as LVMI ≥ 115 g/m^2^ for men and ≥ 95 g/m^2^ for women. ** *p* < 0.05 AUC for comparison with Sokolow–Lyon index in relevant CV risk groups.

**Table 7 jcm-09-01364-t007:** Diagnostic accuracy of electrocardiographic criteria to detect the left ventricular hypertrophy in ECHO using LVMI_BSA_ in the study population according to cardiovascular risk classes at different assumed sensitivities and specificities.

**Study Population**	**Variable**							
	**Cut-Off Point, mm**	**Sensitivity, % ***	**Cut-Off Point, mm**	**Sensitivity, % ****	**Cut-Off Point, mm**	**Specificity, % *****	**Cut-Off Point, mm**	**Specificity, % ******
**All Study Population**								
Sokolow–Lyon index, mm	>30.85	11.59	>27.85	18.84	>13.45	9.23	>16.35	22.41
Cornell index, mm	>18.36	21.74	>15.65	36.23	>8.05	28.83	>9.45	40.53
Lewis index, mm	>12.55	20.29	>8.90	34.78	>−2.70	32.95	>0.65	48.93
**Low-Risk**								
Sokolow–Lyon index, mm	<14.95	17.65	<16.55	23.53	<33.15	10.03	<26.85	26.02
Cornell index, mm	>18.15	23.53	>14.50	29.41	>6.85	27.90	>9.35	48.28
Lewis index, mm	>11.50	29.41	>6.95	52.94	>-0.30	59.25	>1.35	68.03
**Moderate-Risk**								
Sokolow–Lyon index, mm	>29.90	6.67	>27.20	13.33	>13.50	15.08	>14.10	20.63
Cornell index, mm	<7.15	13.33	<8.35	26.67	<15.30	20.63	<13.50	32.54
Lewis index, mm	<−4.00	6.67	<−1.90	13.33	<12.35	11.11	<9.50	20.63
**High and Very-High Risk**								
Sokolow–Lyon index, mm	>29.60	16.13	>25.25	32.26	>15.45	21.97	>17.05	29.55
Cornell index, mm	>18.90	22.58	>16.45	45.16	>8.90	20.45	>10.35	30.30
Lewis index, mm	>15.10	19.35	>11.25	29.03	>−1.00	21.97	>1.70	34.09

* Sensitivity at 90% specificity, ** Sensitivity at 80% specificity; *** Specificity at 90% sensitivity; **** Specificity at 80% sensitivity.

**Table 8 jcm-09-01364-t008:** The comparison of the frequency of exertion dyspnea between individuals with and without left ventricular hypertrophy in low, moderate, high, and very high cardiovascular risk classes.

CV Risk	Frequency of Exertion Dyspnea *		
	No LVH **	LVH **	*p*
**Using ECHO**			
Low risk	44 (14.9)	6 (37.5)	0.028
Moderate risk	30 (28.6)	4 (33.3)	0.743
High risk	14 (19.4)	2 (11.8)	0.727
**Using Sokolow–Lyon Index**			
Low risk	49 (16.3)	1 (4.8)	0.219
Moderate risk	34 (28.3)	1 (50)	0.493
High risk	16 (17.4)	1 (33.3)	0.450
**Using Cornel Index**			
Low risk	48 (15.3)	2 (25.0)	0.361
Moderate risk	35 (29.2)	0 (0.0)	1.000
High risk	17 (18.1)	0 (0.0)	1.000
**Using Lewis Index**			
Low risk	48 (15.2)	2 (40.0)	0.174
Moderate risk	34 (29.6)	1 (14.3)	0.672
High risk	17 (19.1)	0 (0.0)	0.587

The data are shown as n (%); CV: cardiovascular; LVH: left ventricular hypertrophy. The comparisons of categorical variables between subgroups were conducted using the Fisher’s Exact test. * Individuals with asthma and chronic obstructive pulmonary diseases were excluded. ** The left ventricular mass (LVM) index was calculated by the formula LVM/BSA, and the LVH was defined as LVMI ≥ 115 g/m^2^ for men and ≥ 95 g/m^2^ for women.

**Table 9 jcm-09-01364-t009:** Last visit to a doctor or for a diagnostic procedure in patients by CV risk category with respect to the presence of LVH in ECHO *.

		No LVH *	LVH *	*p*
**Low CV Risk**				
Last visit to a doctor or for a diagnostic procedure	during the last month	113 (35.6)	12 (70.6)	0.014
	during the last year	164 (51.7)	5 (29.4)	
	over a year ago or at all	40 (12.6)	0 (0.0)	
**Moderate CV Risk**				
Last visit to a doctor or for a diagnostic procedure	during the last month	55 (44.4)	3 (20.0)	0.001
	during the last year	59 (47.6)	5 (33.3)	
	over a year ago or at all	10 (8.1)	7 (46.7)	
**High CV Risk**				
Last visit to a doctor or for a diagnostic procedure	during the last month	39 (48.8)	9 (45.0)	0.936
	during the last year	34 (42.5)	9 (45.0)	
	over a year ago or at all	7 (8.8)	2 (10.0)	
**Very High CV Risk**				
Last visit to a doctor or for a diagnostic procedure	during the last month	25 (49.0)	7 (63.6)	0.603
	during the last year	22 (43.1)	3 (27.3)	
	over a year ago or at all	4 (7.8)	1 (9.1)	

The data are shown as *n* (%); CV: cardiovascular; LVH: left ventricular hypertrophy. * The left ventricular mass (LVM) index was calculated by the formula LVM/BSA, and the LVH was defined as LVMI ≥ 115 g/m^2^ for men and ≥ 95 g/m^2^ for women.

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
