# Peer review of "ECG Indices Poorly Predict Left Ventricular Hypertrophy and Are Applicable Only in Individuals with Low Cardiovascular Risk"

_jcm, 2020, doi:10.3390/jcm9051364_

Round 1
Reviewer 1 Report
In this paper, Chlabicz et al. provided an analysis of the ECG indices to predict LV hypertrophy in the general population. In a cohort of 676 patients recruited that met inclusion criteria, authors showed that 10% of the population presented LVH. The performance of ECG indices currently used were at the best modest to predict LVH. They concluded that new indices should be provided to improve risk stratification and implement adequate management and treatment for these patients.
1 – It is difficult to clearly understand which kind of analyses was done to assess the determinants of LVH. Authors stated they used linear model but LVH is a dichotomized variable: did you perform logistic model or linear model? Which variable did you used as end-point for this analysis? In the table 4, did you test each variable included in this table individually after adjustment for variables included in the models 1 and 2? The adjustment was not done for renal function; could you add it to your model? More generally, how did you choose the variables included in the models? The used of standardized coefficient should be preferred to help readers interpret this table and these results.
2 – In addition to the analysis of LVH as dichotomized variable, an analysis of the determinants of LVMi should be provided? Did you identify the same determinants?
3 – A relatively important proportion of patients were obese or presented overweight. An indexation by the 2.7 power of height could be of interest.
4 – In addition to the use of the Yuden index to determine the “best” cut-off to predict LVH, could provide cut-off that are more sensible and/or specific? Positive and negative predictive value for all these cut-offs should be added. What was the objective of LVH identification by ECG? Should we favor a less stringent index and then refer patients to echo to confirm LVH?
5 – The analyses, and especially the subgroup analyses, are limited by the number of patients with LVH. These data should be interpreted with caution.
6 – Authors concluded and discussed about obesity markers. However, in your multivariable models, these factors were not associated with LVH. Why did you emphasize these ones?
7 – I am wondering if the patients that you identify with the 3 ECG indices were the same patients? Could you provide an analysis including the 3 indices mixed up to increase the accuracy of the ECG-based identification? We can also argue to build something that could, in addition to ECG indices, include biological/circulating markers (based on your findings, maybe TnT, HbA1c)? This king of analysis could be provided by the authors, and should be of interest for the community.
Minor points.
In the abstract, you presented 717 patients but the study was only conducted on 676 that met inclusion criteria.
Arterial hypertension was abbreviated with HA and AH. Please correct this.
It could be important to clearly state that blood sample has collected following a fasting period
Author Response
Journal of Clinical Medicine
Prof. Karol Kaminski MD PhD FESC
Department of Population Medicine and Civilization Diseases Prevention
Medical University of Bialystok, Poland
fizklin@wp.pl
Re: jcm-777163
Editor-in-Chef
Journal of Clinical Medicine
Prof. Dr. Emmanuel Andrès
24-Apr-2020
Dear Professor Emmanuel Andrès,
I am pleased to resubmit an original research article entitled “ECG indices poorly predict left ventricular hypertrophy and work only in individuals with low cardiovascular risk.” by Malgorzata Chlabicz et al. for consideration for publication in Journal of Clinical Medicine. It was assigned the number jcm-777163. The present study provides evidence on the frequency of left ventricular hypertrophy in general population and on the ability to detect it using current electrocardiographic indicators in terms of a cardiovascular risk level.
Thank you very much for the reviews of the manuscript. We revised our manuscript and did all additional work according to reviewers’ comments and suggestions. The language, style and spelling have been corrected. We have done a number of analyzes suggested by reviewers. Some of them were added to the manuscript, a part enclosed as a supplement. The changed parts of manuscript are colored in red. Please find below detailed answers to comments of the reviewers.
We believe that this manuscript is interesting for the readers of your esteemed Journal of Clinical Medicine, as it improves the knowledge accordingly the frequency of left ventricular hypertrophy in general population and on the ability of detection of LVH using known electrocardiographic indicators in terms of a cardiovascular risk level.
I herby confirm that this manuscript has not been published previously and is not under consideration for publication elsewhere. We declare that we have no conflicts of interest to disclose. Thank you for your consideration.
Kind regards,
Karol Kaminski
Department of Population Medicine and Civilization Diseases Prevention
Medical University of Bialystok, Poland
Reviewer #1
- It is difficult to clearly understand which kind of analyses was done to assess the determinants of LVH. Authors stated they used linear model but LVH is a dichotomized variable: did you perform logistic model or linear model? Which variable did you used as end-point for this analysis? In the table 4, did you test each variable included in this table individually after adjustment for variables included in the models 1 and 2? The adjustment was not done for renal function; could you add it to your model? More generally, how did you choose the variables included in the models? The used of standardized coefficient should be preferred to help readers interpret this table and these results.
In main manuscript we used multivariable logistic regression analysis (we specified it), the end-point was presence of LVH in echocardiography defined.
In Table 4 we added standardized OR for independent variables.
We have done new multivariable logistic regression models (Model 3 and Model 4) without BMI as covariate (Table 5).
We have done a number of analyzes suggested by Reviewers. We used also other calculation of LVMI – dividing LVM by height2.7 Multivariable linear regression analysis was standardized for independent and dependent variables. We used un standardized coefficient and standardized coefficient for each analysis. The end-point in multivariable linear regression analysis was LVMI.
In supplement we enclosed:
Table S1. Results of the left ventricular mass index (calculated by the formula LVMBSA) multivariable linear regression analysis in the study population.
Table S2. Results of the left ventricular mass index (calculated by the formula LVMBSA) multivariable linear regression analysis in the study population.
Table S3. Results of the left ventricular hypertrophy (LVMI calculated by the formula LVMHeight) multivariable logistic regression analysis in the study population.
Table S4. Results of the left ventricular hypertrophy (LVMI calculated by the formula LVMHeight) multivariable logistic regression analysis in the study population.
Table S5. Results of the left ventricular mass LVM index (calculated by the formula LVMHeight ) multivariable linear regression analysis in the study population.
Table S6. Results of the left ventricular mass LVM index (calculated by the formula LVMHeihgt) multivariable linear regression analysis in the study population.
In Model 1 we used variables that have a proven relationship with LVH (age, sex, GFR, BMI), in Model 2 we added parameters that are associated with very- high cardiovacsular risk (history of AH, MI, IHD, PAD, DM, AF, stroke and BP≥140 and/or≥90 mmHg) to find parameters that will be associated with LVH regardless of the CV risk class. We removed BMI from Model 3 and 4 as covariate. In Model 3 we used age, sex, GFR, in Model 4 we added parameters that are associated with very- high cardiovacsular risk (history of AH, MI, IHD, PAD, DM, AF, stroke and BP≥140 and/or≥90 mmHg). The variables that remained associated with LVH in Model 3. were QRS duration, LAVI, hs-TnT, HbA1c, the Cornel index, the Lewis index; in Model 4 the Sokolow-Lyon index remained irrelevant, and we confirmed the relationship between LVH and abdominal obesity regardless of occurrence of CV disease (Table 5). This analysis without BMI as covariate confirmed positive significant correlation between LVMI and BMI, moreover with android-type obesity.
In addition, we performed the multivariable linear regression analysis of LVMIBSA, variables that were positively correlated with LVMIBSA in Model 1. and 2. were very similar to those mentioned earlier (Table S1). In Model 3 after removing BMI as covariate, confirmed the relationship between LVMI and abdominal obesity, but in Model 4 some of them become irrelevant (Table S2).
In our study 36.1% population was overweight and 23.8% were obese. It is known that LVM indexed to BSA underestimates LVH prevalence in obese as well as overweight individuals, hence following the suggestion of the reviewer we calculated LVMI by formula LVM/Height in m2.7. In Table S3 and S4 we showed the multivariable logistic regression analysis, variables that remained associated with LVHHeight in Model 1. and 2.were QRS duration, LAVI, HbA1c, the Cornel index, the Lewis index. In Model 3 and 4 appeared hs-TnT, fasting glucose, 120min glucose, fasting insulin, 120min insulin and indicators of abdominal obesity. Moreover, in the multivariable linear regression analysis of LVMIHeight the relationship with carbohydrate metabolism and abdominal obesity were even more expressed (Table S5-S6).
These analyzes were corrected for renal function, glomerular filtration rate Cockcroft-Gault Equation (GFR) was used in each models.
We added the following paragraphs in the manuscript:
In Model 1 we used variables that have a proven relationship with LVH (age, sex, GFR, BMI), in Model 2 we added parameters that are associated with very-high cardiovacsular risk (history of HA, MI, IHD, PAD, DM, AF, stroke and BP≥140 and/or≥90 mmHg) to find parameters that will be associated with LVH regardless of the CV risk class.
We removed BMI from Model 3 and 4 as covariatein order to reveal the effect of fat tissue excess. The variables that remained associated with LVH in Model 3. were QRS duration, LAVI, hs-TnT, HbA1c, the Cornel index, the Lewis index; in Model 4 the Sokolow-Lyon index remained irrelevant, and we confirmed the relationship between LVH and abdominal obesity regardless of occurrence of CV disease (Table 5).
In addition, we performed the multivariable linear regression analysis of LVMIBSA, variables that were positively correlated with LVMIBSA in Model 1. and 2. were very similar to those found in logistic regression (Table S1). In Model 3 after removing BMI as covariate, we confirmed the relationship between LVMI and abdominal obesity (Table S2).
In our study 36.1% population was overweight and 23.8% obese. It is known that LVM indexed to BSA underestimates LVH prevalence in obese as well as overweight individuals, therefore we calculated LVMI by formula LVM/Height in m2.7. In Table S3 and S4 we showed the multivariable logistic regression analysis, variables that remained associated with LVH using LVMIHeight in Model 1. and 2.were QRS duration, LAVI, HbA1c, the Cornel index, the Lewis index. In Model 3 and 4 appeared hs-TnT, fasting glucose, 120min glucose, fasting insulin, 120min insulin and indicators of abdominal obesity. Moreover, in the multivariable linear regression analysis of LVMIHeight the relationship with carbohydrate metabolism and abdominal obesity were even more pronounced (Table S5-S6).
Additional sentence in the discussion
This phenomenon was even more pronounced in the analysis using LVM indexed to Height in m2.7, which is less dependent on body weight.
Conclusions: Additional analyzes confirm our previous results, that LVH is associated with android-type obesity, LAVI, QRS duration, hs-TnT and HbA1c.
We believe that additional analyzes suggested by the reviewer supported and further strengthened our conclusions.
- In addition to the analysis of LVH as dichotomized variable, an analysis of the determinants of LVMi should be provided? Did you identify the same determinants?
We have done it according to the reviewers suggestions. See point 1.
Table S1-S2, we identified very similar determinants.
- A relatively important proportion of patients were obese or presented overweight. An indexation by the 2.7 power of height could be of interest.
We have done it according to the reviewers suggestions. See point 1. Table S3-S6.
We calculated the left ventricular mass (LVM) index by the formula LVM/m2.7(LVH was defined as LVMI ≥50 g/m2.7 for men and ≥47 g/m2.7 for women), we identified the same determinants in Model 1 and 2. In Model 3 and 4 more parameters of carbohydrate metabolism were significantly related with LVH.
- In addition to the use of the Yuden index to determine the “best” cut-off to predict LVH, could provide cut-off that are more sensible and/or specific? Positive and negative predictive value for all these cut-offs should be added. What was the objective of LVH identification by ECG? Should we favor a less stringent index and then refer patients to echo to confirm LVH?
We added positive and negative predictive values in Table 5.
We made an additional Table 7 (Diagnostic accuracy of electrocardiographic criteria to detect the left ventricular hypertrophy (using LVMIBSA) in study population and according cardiovascular risk classes at different sensitivity and specificity).
The method to screen in pre-clinical stage should be of public health importance. The screening method should be safe and the condition should have an effective treatment. Moreover, higher diagnostic sensitivity than currently presented is in particular demand in population screening. On the other hand we cannot afford significantly lowering specificity because it would mean increased use of resources and psychological and social (time spent, lost income) burden on patients with false positive results.
We added the following sentence to the discussion section:
We confirmed in the current analysis low sensitivity of the Sokolow-Lyon voltage criteria, for all study population were 11.59% or 18.84 % at specificity 90% or 80%, respectively. While the sensitivity of the Cornell voltage criteria were better, for all study population were 21.74% or 36.23% at specificity 90% or 80%, respectively. In previous studies the proposed criteria for ECG diagnosis of LVH had typically high (>90%) specificities with lower sensitivities (6.9-60%) [59], renders the ECG a relatively ineffective method for identification individuals with LVH. Maybe we should consider increasing sensitivity at the expense of specificity or adding new non-ECG parameters that may increase sensitivity without compromising specificity.
- The analyses, and especially the subgroup analyses, are limited by the number of patients with LVH. These data should be interpreted with caution.
We agree with Reviewer. We treat this part of the study as hypothesis generating.
We add following sentence to the limitation section:
We treat subgroup analyses in this study as hypothesis generating, therefore this part should be interpreted with caution.
- Authors concluded and discussed about obesity markers. However, in your multivariable models, these factors were not associated with LVH. Why did you emphasize these ones?
We mentioned obesity because in our analysis LVH is associated with parameters of android-type obesity, and the reason of decreasing sensitivity of Sokolow-Lyon index in the population could be visceral obesity - the increase of epicardial fat resulting in a higher distance between the heart and the skin electrodes on the chest wall. We admit that the latter part is merely a speculation.
In multivariable logistic regression analysis we used BMI as covariate because it has established relationship with LVH.
We made an additional analyses (Table 5, Table S2, Table S4, Table S6) using adjustment for age, sex, GFR in Model 3; age, sex, GFR, history of AH, MI, IHD, PAD, DM, AF, stroke and BP≥140 and/or≥90 mmHg in Model 4 and confirmed positive significant association of android-type obesity with LVMI and LVH.
- I am wondering if the patients that you identify with the 3 ECG indices were the same patients? Could you provide an analysis including the 3 indices mixed up to increase the accuracy of the ECG-based identification? We can also argue to build something that could, in addition to ECG indices, include biological/circulating markers (based on your findings, maybe TnT, HbA1c)? This king of analysis could be provided by the authors, and should be of interest for the community.
We agree with the reviewer. Our work suggests that adds these parameters may be relevant. Now we are looking for a validation cohort. This a new task, impossible to perform at the time required by the Editor (7 days).
- In the abstract, you presented 717 patients but the study was only conducted on 676 that met inclusion criteria.
We changed following sentence in the abstract :
676 volunteers were included.
- Arterial hypertension was abbreviated with HA and AH. Please correct this.
We corrected this.
- It could be important to clearly state that blood sample has collected following a fasting period.
Yes, the peripheral intravenous fasting blood samples were collected at the time of visit, which always took place in the morning.
This was describe in data collection and assays section (verse 76-78).
Reviewer 2 Report
The researchers present a comprehensive study examining the diagnostic value of ECG features for detecting LVH in patients with various CV risks in the general population. Overall, the authors did an excellent job in the methodology of the work and presentation of their findings. Furthermore, the article is well written with only minor grammatical errors.
The authors demonstrate perhaps expected, yet needed, results. They demonstrated that current ECG criteria for diagnosing LVH do not have great predictive power and may vary amongst CV risk level. More specifically, the Sokolow-Lyon criteria are noted to be unreliable, while the Cornell voltage and Lewis voltage criteria are useful in low CV risk patients. This does raise an important concern with current ECG features to identify LVH and the need for more accurate means to detect LVH. The ECG remains a noninvasive, cost-effective, and widely available mode that should be explored further in LVH detection for such reasons. The advancement of artificial intelligence-enabled ECG algorithms may also provide help in increasing sensitivity of detection in the general population. Despite, the limited number of LVH patients and use of ECHO to confirm LVH (instead of cardiac MRI), it is clear that detecting these patients in the general population has benefits with available known risk reduction measures. This work helps to pave the way for further research by demonstrating the limited value of current ECG features to detect LVH.
Author Response
Journal of Clinical Medicine
Prof. Karol Kaminski MD PhD FESC
Department of Population Medicine and Civilization Diseases Prevention
Medical University of Bialystok, Poland
fizklin@wp.pl
Re: jcm-777163
Editor-in-Chef
Journal of Clinical Medicine
Prof. Dr. Emmanuel Andrès
24-Apr-2020
Dear Professor Emmanuel Andrès,
I am pleased to resubmit an original research article entitled “ECG indices poorly predict left ventricular hypertrophy and work only in individuals with low cardiovascular risk.” by Malgorzata Chlabicz et al. for consideration for publication in Journal of Clinical Medicine. It was assigned the number jcm-777163. The present study provides evidence on the frequency of left ventricular hypertrophy in general population and on the ability to detect it using current electrocardiographic indicators in terms of a cardiovascular risk level.
Thank you very much for the reviews of the manuscript. We revised our manuscript and did all additional work according to reviewers’ comments and suggestions. The language, style and spelling have been corrected. We have done a number of analyzes suggested by reviewers. Some of them were added to the manuscript, a part enclosed as a supplement. The changed parts of manuscript are colored in red. Please find below detailed answers to comments of the reviewers.
We believe that this manuscript is interesting for the readers of your esteemed Journal of Clinical Medicine, as it improves the knowledge accordingly the frequency of left ventricular hypertrophy in general population and on the ability of detection of LVH using known electrocardiographic indicators in terms of a cardiovascular risk level.
I herby confirm that this manuscript has not been published previously and is not under consideration for publication elsewhere. We declare that we have no conflicts of interest to disclose. Thank you for your consideration.
Kind regards,
Karol Kaminski
Department of Population Medicine and Civilization Diseases Prevention
Medical University of Bialystok, Poland
Reviewer #2
Thank you very much for the reviews of the manuscript.
Round 2
Reviewer 1 Report
Authors have addressed my comments.